# METAPHYS: FEW-SHOT ADAPTATION FOR NON-CONTACT PHYSIOLOGICAL MEASUREMENT

## ABSTRACT

There are large individual differences in physiological processes, making designing personalized health sensing algorithms challenging. Existing machine learning systems struggle to generalize well to unseen subjects or contexts, especially in video-based physiological measurement. Although fine-tuning for a user might address this issue, it is difficult to collect large sets of training data for specific individuals because supervised algorithms require medical-grade sensors for generating the training target. Therefore, learning personalized or customized models from a small number of unlabeled samples is very attractive as it would allow fast calibrations. In this paper, we present a novel meta-learning approach called MetaPhys for learning personalized cardiac signals from 18-seconds of video data. MetaPhys works in both supervised and unsupervised manners. We evaluate our proposed approach on two benchmark datasets and demonstrate superior performance in cross-dataset evaluation with substantial reductions (42% to 44%) in errors compared with state-of-the-art approaches. Visualization of attention maps and ablation experiments reveal how the model adapts to each subject and why our proposed approach leads to these improvements. We have also demonstrated our proposed method significantly helps reduce the bias in skin type.

## 1 INTRODUCTION

The importance of scalable health sensing has been acutely highlighted during the SARS-CoV-2 (COVID-19) pandemic. The virus has been linked to increased risk of myocarditis and other serious cardiac (heart) conditions (Puntmann et al., 2020). Contact sensors (electrocardiograms, oximeters) are the current gold-standard for measurement of heart function. However, these devices are still not ubiquitously available, especially in low-resource settings. The development of video-based contactless sensing of vital signs presents an opportunity for highly scalable physiological monitoring. Furthermore, in clinical settings non-contact sensing could reduce the risk of infection for vulnerable patients (e.g., infants and elderly) and the discomfort caused to them (Villarroel et al., 2019).

While there are compelling advantages of camera-based sensing, the approach also presents unsolved challenges. The use of ambient illumination means camera-based measurement is sensitive to **environmental differences** in the intensity and composition of the incident light. Camera **sensor differences** mean that hardware can differ in sensitivity across the frequency spectrum. People (the subjects) exhibit large **individual differences** in appearance (e.g., skin type, facial hair) and physiology (e.g, pulse dynamics). Finally, **contextual differences** mean that motions in a video at test time might be different from those seen in the training data. One specific example is that there exists biases in performance across skin types Nowara et al. (2020). This problem is not isolated to physiological measurement as studies have found systematic biases in facial gender classification, with error rates up to 7x higher on women than men and poorer performance on people with darker skin types (Buolamwini & Gebru, 2018). Moreover, there are several challenges in collecting large corpora of high-quality physiological data: 1) recruiting and instrumenting participants is often expensive and requires advanced technical expertise, 2) the data can reveal the identity of the subjects and/or sensitive health information meaning it is difficult for researchers to share such datasets. Therefore, training supervised models that generalize well across environments and subjects is challenging. For these reasons we observe that performance on cross-dataset evaluation is significantly worse than within-dataset evaluation using current state-of-the-art methods (Chen & McDuff, 2018; Liu et al., 2020).

Calibration of consumer health sensors is often performed in a clinic, where a clinician will collect readings from a high-end sensor to calibrate a consumer-level device the patient owns. The reason for this is partly due to the variability within readings from consumer devices across different individuals. Ideally, we would be able to train a personalized model for each individual; however, standard supervised learning training schemes require large amounts of labeled data. Getting enough physiological training data of each individual is difficult because it requires using medical-grade devices to provide reliable labels. Being able to generate a personalized model from a small amount of training samples would enable customization based on a few seconds or minutes of video captured while visiting a clinic where people have access to a gold-standard device. Furthermore, if this process could be achieved without even the need for these devices (i.e., in an unsupervised manner), that would have even greater impact. Finally, combining remote physiological measurement with telehealth could provide patients' vital signs for clinicians during remote diagnosis. Given that requests for telehealth appointments have increased more than 10x during COVID-19, and that this is expected to continue into the future (Smith et al., 2020), robust personalized models are of growing importance.

Meta-learning, or learning to learn, has been extensively studied in the past few years (Hospedales et al., 2020). Instead of learning a specific generalized mapping, the goal of meta-learning is to design a model that can adapt to a new task or context with a small amount of data. Due to the inherent ability for fast adaption, meta-learning is a good candidate strategy for building personalized models (e.g., personalization in dialogue and video retargeting (Madotto et al., 2019; Lee et al., 2019).) However, we argue that meta learning is underused in healthcare where clinicians can quickly adapt their clinical knowledge to different patients. The goal of this work is to develop a meta-learning based personalization framework in remote physiological measurement with a limited amount of data from an unseen individual (task) to mimic how a clinician manually calibrates sensor readings for a specific patient. When meta-learning is applied to remote physiological measurement, there are two kinds of scenarios: 1) supervised adaptation with few samples of labeled data from a clinical grade sensor and 2) unsupervised adaptation with unlabeled data. We hypothesize that supervised adaptation is more likely to yield a robust personalized model with only a few labels, while unsupervised adaptation may personalize the model less effectively but with much lower effort and complexity.

In this paper, we propose a novel meta-learning approach to address the aforementioned challenges called MetaPhys. Our contributions are: 1) A meta-learning based deep neural framework, supporting both **supervised and unsupervised** few-shot adaptation, for camera-based vital sign measurement; 2) A systematic cross-dataset evaluation showing that our system considerably outperforms the state-of-the-art (42% to 52% reduction in heart rate error); 3) To perform an ablation experiment, freezing weights in the temporal and appearance branches to test sensitivity during adaptation; 4) An analysis of performance for subjects with different skin types. Our code, example models, and video results can be found on our github page.[1]

## 2 BACKGROUND

**Video-Based Physiological Measurement:** Video-based physiological measurement is a growing interdisciplinary domain that leverages ubiquitous imaging devices (e.g., webcams, smartphones' cameras) to measure vital signs and other physiological processes. Early work established that changes in light reflected from the body could be used to capture subtle variations blood volume and motion related to the photoplethysmogram (PPG) (Takano & Ohta, 2007; Verkruysse et al., 2008) and ballistocardiogram (BCG) (Balakrishnan et al., 2013), respectively. Video analysis enables non-contact, spatial and temporal measurement of arterial and peripheral pulsations and allows for magnification of theses signals (Wu et al., 2012), which may help with examination (e.g., (Abnousi et al., 2019)). Based on the PPG and BCG signal, heart rate can be extracted (Poh et al., 2010b; Balakrishnan et al., 2013).

However, the relationship between pixels and underlying physiological changes in a video is complex and neural models have shown strong performance compared to source separation techniques (Chen & McDuff, 2018; Yu et al., 2019; Zhan et al., 2020). Conventional supervised learning requires a large amount of training data to produce a generalized model. However, obtaining a large body of physiological and facial data is complicated and expensive. Current public datasets have limited

---

[1]https://github.com/anonymous0paper/MetaPhys

numbers of subjects and diversity in regards of appearance (including skin type), camera sensors, environmental conditions and subject motions. Therefore, if the subject of interest is not in the training data or the video is otherwise different, performance can be considerably degraded, a result that is not acceptable for a physiological sensor.

Lee et al. (Lee et al., 2020) recognized the potential for meta-learning applied to imaging-based cardiac pulse measurement. Their method (Meta-rPPG) focuses on using unsupervised meta-learning and a LSTM encoder-decoder architecture which to our knowledge was not validated in previous work. Instead, our proposed meta-learning framework is built on top of a state-of-the-art on-device network (Liu et al., 2020) and aims to explore the potential of both supervised and unsupervised on-device personalized meta-learning. Meta-rPPG uses a synthetic gradient generator and a prototypical distance minimizer to perform transductive inference to enable unsupervised meta-learning. This learning mechanism requires a number of rather complex steps. We propose a relatively simpler mechanism that is physiologically and optically grounded (Wang et al., 2016; Liu et al., 2020) and achieves greater accuracy.

**Meta-Learning and Person Specific Models:** The ability to learn from a small number of samples or observations is often used as an example of the unique capabilities of human intelligence. However, machine learning systems are often brittle in a similar context. Meta-learning approaches tackle this problem by creating a general learner that is able to adapt to a new task with a small number of training samples, inspired by how humans can often master a new skill without many observations (Hospedales et al., 2020). However, most of the previous work in meta-learning focuses on supervised vision problems (Zoph et al., 2018; Snell et al., 2017) and in the computer vision literature has mainly been applied to image analysis (Vinyals et al., 2016; Li et al., 2017). Supervised regression in video settings has received less attention. One of few examples is object or face tracking (Choi et al., 2019; Park & Berg, 2018). In these tasks, the learner needs to adapt to the individual differences in appearance of the target and then track it across frames, even if the appearance changes considerably over time in the video. Choi et al. (2019) present a matching network architecture providing the meta-learner with information in the form of loss gradients obtained using the training samples.

The property of fast adaptation makes meta-learning a good candidate for personalizing models, it has been used in various applications such as dialogue agents (Madotto et al., 2019), gaze estimation (He et al., 2019), sleep stage classification (Banluesombatkul et al., 2020), activity recognition (Gong et al., 2019), and video retargeting (Lee et al., 2019). For example, Banluesombatkul et al. proposed a MAML-based meta-learning system to perform fast adaption of a sleep stage classification model using biosignals (Banluesombatkul et al., 2020). More recently, MetaPix (Lee et al., 2019) leveraged a meta-learning training schema with a small amount of video to adapt a universal generator to a particular background and human in the problem of video retargeting. Similarly, our proposed meta-learning framework is also capable of personalizing a universal remote physiological model to a new person or an environmental setting.

## 3    METHOD

### 3.1    PHYSIOLOGICAL META-LEARNING

In camera-based cardiac measurement, the goal is to separate pixel changes due to volumetric variations in blood and pulsatile motions from other variations that are not related to the pulse signal. Examples of "noise" in this context that might impact the performance on the task include: changes in the environment (illumination) and changes in appearance of the subject and motions (e.g., facial expressions, rigid head motions). A model trained within a traditional supervised learning regime might perform well if illumination, non-pulsatile motions, and appearances in the test set are similar to those in the training set. However, empirical evidence shows that performance usually significantly degrades from one dataset to another, suggesting that traditional training is likely to overfit to the training set to some extent (Chen & McDuff, 2018). Therefore, to achieve state-of-the-art performance in remote physiological measurement on cross-dataset evaluation, the system should have: 1) a good initial representation of the mapping from the raw video data to pulse signal, and 2) a strategy for adapting to unseen individuals and environments.

To achieve this, we propose a system called MetaPhys, an adaptable meta-learning based on-device framework aimed at efficient and personalized remote physiological sensing. MetaPhys uses a

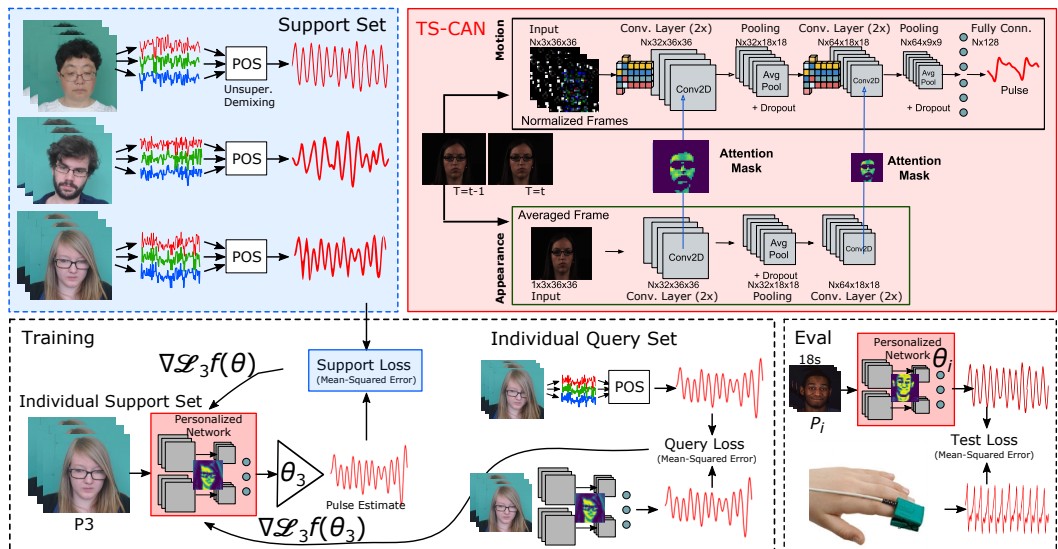

Figure 1: We present MetaPhys, an approach for few-shot unsupervised adaptation for personalized camera-based physiological measurement models.

pretrained convolutional attention network as the backbone (described below) and leverages a novel personalized meta-learning schema to overcome the aforementioned limitations. We adopt Model-Agnostic Meta-Learning (MAML) (Finn et al., 2017) as our personalized parameter update schema. MAML produces a general initialization as the starting point for fast adaptation to a diverse set of unseen tasks with only a few training samples. However, applying MAML to the task of camera-based physiological measurement has differences to many previously explored meta-learning problems. Existing meta-learning approaches are often evaluated on classification or some toy regression tasks due to the lack of regression benchmark datasets (Hospedales et al., 2020). Our problem is a non-trivial vision-based regression task due to the subtle nature of the underlying physiological signal. Algorithm 1 outlines the training process for MetaPhys, we first pretrain the backbone network to get an initial spatial-temporal representation. Then we treat each individual as a task $\tau_i$. During the training, we split the data into a support set ($K$ video frames) and a query set ($K'$ video frames) for each individual (task). The support set is used to update the task's parameters and yield a personalized model $\theta_i$. The query set is used to assess the effectiveness of the personalized model and further update the global initialization $\theta$ to make future adaptation better. A robust personalized model $\theta_i$ aims to provide a more accurate attention mask to the corresponding motion branch and to preform precise physiological measurement for the target individual as well as the target's environment. During the testing stage, MetaPhys has the updated global initialization $\hat{\theta}$, and can generate $\hat{\theta}_i$ for each test individual (task) by optimizing the test support set as $\hat{\theta_{\tau_i}} \leftarrow \hat{\theta} - \alpha\nabla_{\hat{\theta}}\mathcal{L}_{\tau_i}f(\hat{\theta})$. With this training and testing schema, the robust global initialization $\hat{\theta}$ generated from MetaPhys not only leverages the pretrained representation but also learns how to adapt to individual and environmental noise quickly.

## 3.2 Spatial and Temporal Model Architecture Backbone

Our ultimate goal is a computationally efficient on-device meta-learning framework that offers inference at 150 fps. Therefore, we adopt the state-of-the-art architecture (TS-CAN) (Liu et al., 2020) for remote cardiopulmonary monitoring. TS-CAN is an end-to-end neural architecture with appearance and motion branches. The inputs are video frames and the output is the first-derivative of the pulse estimate. Tensor shifting modules (TSM) (Lin et al., 2019) are used that shift frames along the temporal axis allowing for information exchange across time. This helps capture temporal dependencies beyond consecutive frames. The appearance branch and attention mechanism help guide the motion branch to focus on regions with high pulsatile signal (e.g., skin) instead of others (e.g.,

clothes, hair) (see Fig. 1). However, we discover empirically that this network does not necessarily generalize well across datasets with differences in subjects, lighting, backgrounds and motions (see Table 1). One of the main challenges when employing TS-CAN is that the appearance branch may not generate an accurate mask while testing on unseen subjects or environments because of the differences in appearance of skin pixels. Without a good attention mask, motions from other sources are likely to be given more weight, thus damaging the quality of our physiological estimate.

### 3.3 SUPERVISED OR UNSUPERVISED LEARNING

We explore both supervised and unsupervised training regimes for MetaPhys. Supervised personalization may be suitable in clinical settings that require highly precise adaptation and where there is access to reference devices. Unsupervised personalization may be preferable for consumer measurement when convenience and scalability is of a greater priority and calibration with a clinical grade device might be difficult.

For the supervised version of MetaPhys we use the gold standard reference signal from a finger PPG or blood pressure wave (BPW) to train the meta-learner and perform few-shot adaptation when testing. In contrast to the supervised version, in the unsupervised case we use pseudo labels during the training of the MetaPhys meta-learner and parameter updates rather than the ground-truth signal from the medical-grade devices. We use a physiologically-based unsupervised remote physiological measurement model to generate pseudo pulse signal estimates without relying on gold standard measurements. More specifically, we leverage the Plane-Orthogonal-to-Skin (POS) (Wang et al., 2016) method, which is the current state-of-the-art for demixing in this context. POS calculates a projection plane orthogonal to the skin-tone, derived based on optical and physiological principles, that is then used for pulse extraction. In details, POS can be summarized into four steps: 1) spatial averaging each frame, 2) temporal normalization within a certain window size, 3) applying a fixed matrix projection to offset the specular reflections and other noise, 4) band-pass filtering.

We observe that even though our unsupervised model uses the POS signal for meta-training, Meta-Phys's performance significantly outperforms POS once trained. As Algorithm 1 illustrates, the pseudo label generator $G$ produces pseudo labels for both $K$ support frames and $K'$ query frames for adaptation and parameter updates. We used pseudo labels for the query set ($K'$) in training, as we observed similar empirical results in preliminary testing whether we used pseudo labels or ground-truth labels.

---

**Algorithm 1** MetaPhys: Meta-learning for physiological signal personalization

---

**Require:** $S$: Subject-wise video data
**Require:** A batch of personalized tasks $\tau$ where each task $\tau_i$ contains N data points from $S_i$
**Require:** A pseudo label generator $G$ for unsupervised meta-learning
 1: $\theta \leftarrow$ **Pre-training** TS-CAN on AFRL dataset
 2: **for each** $\tau_i \in \tau$ **do**
 3:     **if** Supervised **then**
 4:         $K \leftarrow$ Sample $K$ support frames from videos of $\tau_i$ with ground truth labels
 5:         $K' \leftarrow$ Sample $K'$ query frames from videos of $\tau_i$ with ground truth labels
 6:     **else**
 7:         $K \leftarrow$ Sample $K$ support frames from videos of $\tau_i$ with pseudo labels from $G$
 8:         $K' \leftarrow$ Sample $K'$ query frames from videos of $\tau_i$ with pseudo labels from $G$
 9:     **end if**
10:     $\theta_{\tau_i} \leftarrow \theta - \alpha \nabla_\theta \mathcal{L}_{\tau_i} f(K, \theta)$, Update the personalized params. based on indiv. support loss
11: **end for**
12: $\hat{\theta} \leftarrow \theta - \beta \nabla_\theta \sum_{\tau_i} \mathcal{L}_{\tau_i} f(K'_{\tau_i}, \theta_{\tau_i})$, Update the global params. based on individuals' query loss

---

## 4 EXPERIMENTS

### 4.1 DATASETS

**AFRL (Estepp et al., 2014):** 300 videos of 25 participants (17 males) were recorded at 658x492 resolution and 30 fps. Pulse measurements were recorded via a contact reflectance PPG sensor

Table 1: Pulse Measurement on the MMSE-HR and UBFC datasets.

| Method | Train / Test Datasets | MAE | RMSE | SNR | $\rho$ |
|---|---|---|---|---|---|
| Pretrain + Unsupervised **MetaPhys** | (AFRL & UBFC) / MMSE | **1.87** | **3.12** | 5.04 | **0.89** |
| Pretrain + Supervised **MetaPhys** | (AFRL & UBFC) / MMSE | 2.98 | 4.86 | 3.81 | 0.72 |
| **MetaPhys** (No pretrain) | (AFRL & UBFC) / MMSE | 3.67 | 5.50 | 2.41 | 0.70 |
| Supervised Pretrain + FT on Test Support Set | (AFRL & UBFC) / MMSE | 4.05 | 5.68 | 2.76 | 0.76 |
| Supervised Pretrain (Liu et al., 2020) | (AFRL & UBFC) / MMSE | 3.78 | 5.75 | 2.67 | 0.77 |
| (Unsupervised) CHROM (De Haan & Jeanne, 2013) | None / MMSE | 3.2 | 5.71 | 5.42 | 0.75 |
| (Unsupervised) POS (Wang et al., 2016) | None / MMSE | 3.98 | 6.66 | **5.74** | 0.67 |
| (Unsupervised) ICA (Poh et al., 2010a) | None / MMSE | 4.12 | 6.46 | 6.09 | 0.67 |

| Method | Train / Test Datasets | MAE | RMSE | SNR | $\rho$ |
|---|---|---|---|---|---|
| Pretrain + Unsupervised **MetaPhys** | (AFRL & MMSE) / UBFC | 2.46 | 3.12 | **4.28** | **0.96** |
| Pretrain + Supervised **MetaPhys** | (AFRL & MMSE) / UBFC | **1.90** | **2.62** | 3.84 | 0.96 |
| **MetaPhys** (No pretrain) | (AFRL & MMSE) / UBFC | 3.80 | 5.32 | 0.13 | 0.84 |
| Supervised Pretrain + FT on Test Support Set | (AFRL & MMSE) / UBFC | 6.26 | 7.37 | -0.23 | 0.72 |
| (Unsupervised) Meta-rPPG (Lee et al., 2020) | Self-Collected / UBFC | 5.97 | 7.42 | - | 0.53 |
| Supervised Pretrain (Liu et al., 2020) | (AFRL & MMSE) / UBFC | 4.42 | 6.13 | 1.87 | 0.79 |
| (Unsupervised) POS (Wang et al., 2016) | None / UBFC | 6.44 | 9.48 | 0.55 | 0.66 |
| (Unsupervised) CHROM (De Haan & Jeanne, 2013) | None / UBFC | 7.31 | 9.85 | 0.93 | 0.57 |
| (Unsupervised) ICA (Poh et al., 2010a) | None / UBFC | 10.2 | 14.4 | -0.19 | 0.50 |

MAE = Mean Absolute Error, RMSE = Root Mean Squared Error, $\rho$ = Pearson Correlation, SNR = BVP Signal-to-Noise Ratio.

and used for training. Electrocardiograms (ECG) were recorded for evaluating performance. Each participant was recorded six times with increasing head motion in each task (10 degrees/second, 20 degrees/second, 30 degrees/second). The participants were asked to sit still for the first two tasks and perform three motion tasks rotating their head about the vertical axis.

**UBFC (Bobbia et al., 2019):** 42 videos of 42 participants were recorded at 640x480 resolution and 30 fps in uncompressed 8-bit RGB format. A CMS50E transmissive pulse oximeter was used to obtain the ground truth PPG data. All the experiments were conducted indoors with different sunlight and indoor illumination. Participants were also asked to play time sensitive mathematical games to augment the heart rate during the data collection.

**MMSE (Zhang et al., 2016):** 102 videos of 40 participants were recorded at 1040x1392 resolution and 25 fps. A blood pressure wave signal was measured at 1000 fps as the gold standard. The blood pressure wave was used as the training signal for this data as a PPG signal was not available. The distribution of skin types based on the Fitzpatrick scale Fitzpatrick (1988) is: II=8, III=11, IV=17, V+VI=4.

## 4.2 Implementation Details

MetaPhys was implemented in PyTorch (Paszke et al., 2019), and all the experiments were conducted on a Nvidia 2080Ti GPU. We first implemented the backbone network (TS-CAN) and modified it to use a window size of 20 frames (rather than 10) because we empirically observed a larger window size led to better overall performance. Then, we implemented MetaPhys based on a gradient computation framework called higher (Grefenstette et al., 2019). Compared with most previous meta-learning studies that were trained and evaluated on a single dataset (e.g., miniimagenet (Vinyals et al., 2016)), we used three datasets to perform pretraining and cross-dataset training and evaluation. Our backbone was pretrained on the AFRL dataset, and the training (described in Algorithm 1) and evaluation of our meta-learner were performed with the UBFC and MMSE datasets. We picked the size of the support set ($K$) for personalization to be 540 video frames for each individual. For a 30 fps video recording this equates to an 18-second recording which is a reasonably short calibration period. During the meta training and adaptation, we used an Adam optimizer (Kingma & Ba, 2014) with an outer learning rate ($\beta$) of 0.001 and a stocastic gradient descent (SGD) optimizer with an inner learning rate ($\theta$) of 0.005. We trained the meta-learner for 10 epochs, and performed one step adaptation (i.e., gradient descent).

As baselines, we implemented traditional supervised training (TS-CAN) on AFRL and evaluated on MMSE and UBFC. A conventional fine tuning with the support set and testing on the query set was implemented as our adaptation baseline. To assure a fair comparison across all experiments, we forced the test data (test query set) to remain the same within each task. We also implemented

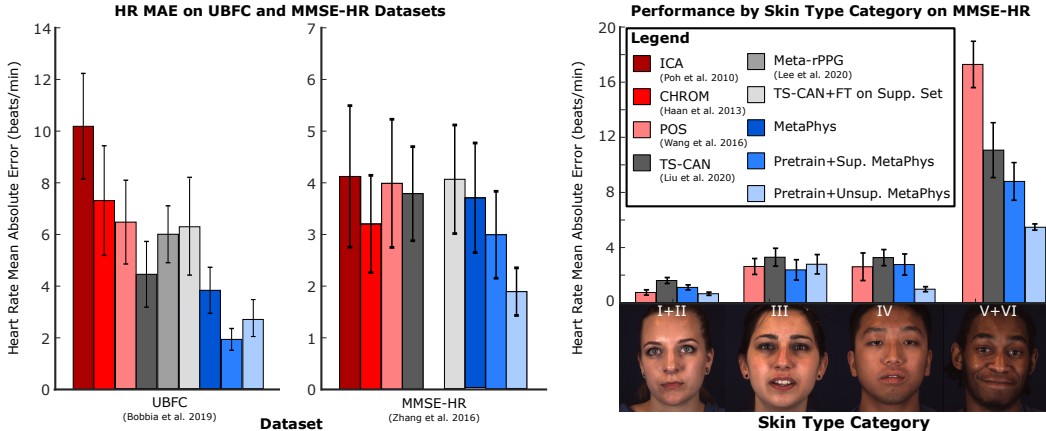

Figure 2: Left) MAE in HR estimates (12-second windows) for the UBFC and MMSE-HR datasets. Right) MAE in HR estimates by skin type on the MMSE-HR dataset. Standard error bars shown.

three established unsupervised algorithms (CHROM, POS, ICA) using iPhys-Toolbox (McDuff & Blackford, 2019). We applied post-processing to the outputs of all the methods in the same way. We first divided the remainder of the recordings for each participant into 360-frame windows (approximately 12 seconds), with no overlap, and applied a 2nd-order butterworth filter with a cutoff frequency of 0.75 and 2.5 Hz (these represent a realistic range of heart rates we would expect for adults). We then computed four metrics for each window: mean absolute error (MAE), root mean squared error (RMSE), signal-to-noise ratio (SNR) and correlation ($\rho$) in heart-rate estimations. Unlike most prior work which evaluated performance on whole videos (often 30 or 60 seconds worth of data), we perform evaluation on 12 second sequences which is considerably more challenging as the model has much less information for inference.

## 5 RESULTS AND DISCUSSION

**Comparison with the State-of-the-Art:** For the MMSE dataset, our proposed supervised and unsupervised MetaPhys with pretraining outperformed the state-of-the-art results by 7% and 42% in MAE, respectively (see Table 1). On the UBFC dataset, supervised and unsupervised MetaPhys with pretraining showed even greater benefits reducing error by 57% and 44% compared to the previous state-of-the-art, respectively. Meta-learning alone is not as effective as meta-learning using weights initialized in a pretraining stage (19% and 50% improvements in MMSE and UBFC). We also compared our methods against the only other meta-learning based method (Meta-rPPG) where we reduced the MAE by 68%. Furthermore, we compared MetaPhys against the traditional personalization method (fine-tuning), and our approach gained a 54% and a 61% improvements in terms of MAE on MMSE and UBFC, respectively. We also evaluated data size of 6s, 12s and 18s for support set during meta training and testing, and the results showed training with 18s (RMSE: 3.12) outperformed 6s (RMSE: 5.43) and 12s (RMSE: 5.53) in the MMSE dataset. A similar trend also has been observed on the UBFC dataset (RMSE of 18s: 3.12, RMSE of 12s: 4.48, RMSE of 6s: 3.46).

**Unsupervised vs. Supervised Adaptation:** Next, we examine the difference between using a supervised and unsupervised training regime in MetaPhys. For UBFC, the *supervised* model (MAE=1.90 BPM), outperformed the *unsupervised* model (MAE=2.46 BPM). Whereas, for the MMSE dataset the *unsupervised* model (MAE=1.87 BMP) outperformed the *supervised* model (MAE=2.98 BMP). The fact that the unsupervised model achieves broadly comparable results to the supervised model is surprising and encouraging because there are many applications where unsupervised adaptation would be more convenient and efficient (e.g., calibrating a heart rate measurement app on a smartphone without needing a reference device). We also observe that the unsupervised model, even though it used the POS signal as training input, significantly outperforms POS on both datasets, suggesting MetaPhys is able to form a better representation.

**Visualizing Adaption:** To help us understand why MetaPhys outperforms the state-of-the-art models we visualized the attention masks for different subjects. In Fig. 3-A, we compare the attention masks

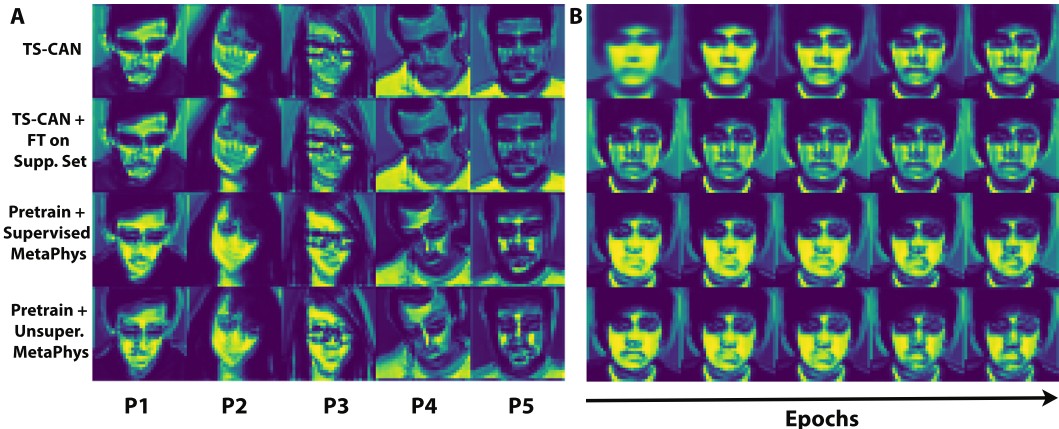

Figure 3: (A) An illustration comparing the attention masks of five subjects. The masks were generated in 4 training schemes: 1) traditional supervised training (TS-CAN), 2)TS-CAN with fine tuning, 3) supervised MetaPhys and 4) unsupervised MetaPhys. (B) An illustration comparing the attention masks in the learning progress from four training schemes.

from the appearance branch of TS-CAN based on four training schemes which are: 1) supervised training with TS-CAN, 2) pretraining TS-CAN on AFRL and then fine tuning TS-CAN on the support set used for the meta-learning experiments, 3) pretraining on AFRL and supervised MetaPhys training, 4) pretraining on AFRL and unsupervised MetaPhys training. The differences are subtle, but on inspection we can notice that MetaPhys leads to masks that put higher weight on regions with greater pulsatile signal (e.g., forehead and cheeks) and less weight on less important regions (e.g., t-shirt - see P5 as an example). In Fig. 3-B, we visualize the progression of learning for the four different methods. Again the changes during learning are subtle, but the traditional supervised methods seem more likely to overfit even over a relatively small number of epochs meaning that the attention to important regions of the face is not as high as with the meta-learning approaches, presumably because the traditional supervised learning has to capture a more generic model which is not well adapted to any one specific individual.

**Freezing Appearance vs. Motion:** We questioned whether the adaptation of the appearance mask was the main or sole reason for the improvements provided by MetaPhys. To test this, we froze the weights in the motion branch of the TS-CAN during the meta-training stage and only updated weights in the appearance branch. From the results of these experiments, we observe that there is a 20% increase in MAE, indicating that MetaPhys not only noticeably improves the quality of the attention mask, but also learns additional temporal dynamics specific to an individual's pulse waveform.

**Robustness to Skin Type:** Our motivation for adopting a meta-learning approach is to improve generalization. One challenge with photoplethysmography methods is their sensitivity to skin type. A larger melanin concentration in people with darker skin leads to higher light absorption compared to lighter skin types (Nowara et al., 2020), thus reducing the reflectance signal to noise ratio. Fig. 2 shows a bar plot of the MAE in heart rate estimates by skin type (we group types I+II and V+VI as there were relatively few subjects in these categories). Both the AFRL and UBFC datasets are heavily skewed towards lighter Caucasian skin type categories. Therefore supervised methods trained on these datasets (e.g., TS-CAN) tend to overfit and not perform well on other skin types. Entirely unsupervised baselines do not perform any better, possibly because they were mostly designed and validated with lighter skin type data as well. While the highest errors for unsupervised MetaPhys still come in the darkest skin type categories, the reduction in error for types V+VI is considerable (68% compared to POS, 50% compared to TS-CAN). We are encouraged that these results are a step towards more consistent performance across people of different appearances.

**Limitations:** There is a trend of performance degradation when the skin type gets darker. We acknowledge this limitation and plan to use resampling to help address this bias in future. Both the MMSE and UBFC datasets have somewhat limited head motion and future work will investigate whether meta-learning can help with generalization to other motion conditions.

## 6 CONCLUSIONS

We present a novel unsupervised few-shot adaptation framework for non-contact physiological measurement called MetaPhys. Our proposed method substantially improves on the state-of-the-art and the performance on various skin types, and we also reveal why and how our method achieved such improvement.

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

# A APPENDIX

## A.1 EVALUATION METRICS

**Mean Absolute Error (MAE):** The MAE between our model estimates and the gold-standard heart rates from the contact sensor measurements were calculated as follows for each 12-second time window:

$$MAE = \frac{1}{T}\sum_{i=1}^{T}|HR_i - HR_i'| \tag{1}$$

**Root Mean Squared Error (RMSE):** The RMSE between our model estimates and the gold-standard heart rate from the contact sensor measurements were calculatd as follows for each 12-second time window:

$$RMSE = \sqrt{\frac{i=1}{T}\sum_{1}^{T}(HR_i - HR_i')^2} \tag{2}$$

In both cases, HR is the gold-standard heart rate and HR' is the estimated heart rate from the video respectively. The gold-standard HR frequency was determined from the calculated from gold-standard PPG signal (UBFC dataset) or blood pressure wave (MMSE dataset).

We also compute the Pearson correlation between the estimated heart rates and the gold-standard heart rates from the contact sensor measurements across all the subjects.

**Signal-to-Noise Ratios (SNR):** We calculate blood volume pulse signal-to-noise ratios (SNR) (De Haan & Jeanne, 2013). This captures the signal quality of the recovered pulse estimates without penalizing heart rate estimates that are slightly inaccurate. The gold-standard HR frequency was determined from the gold-standard PPG waveform (UBFC dataset) or blood pressure wave (MMSE dataset).

$$SNR = 10\log_{10}\left(\frac{\sum_{f=30}^{240}((U_t(f)\hat{S}(f))^2}{\sum_{f=30}^{240}(1 - U_t(f))\hat{S}(f))^2)}\right) \tag{3}$$

where $\hat{S}$ is the power spectrum of the BVP signal (S), $f$ is the frequency (in BPM), HR is the heart rate computed from the gold-standard device and $U_t(f)$ is a binary template that is one for the heart rate region from HR-6 BPM to HR+6BPM and its first harmonic region from 2*HR-12BPM to 2*HR+12BPM, and 0 elsewhere.

## A.2 CODE AND ADDITIONAL RESULTS:

Available at: `https://github.com/anonymous0paper/MetaPhys`.

