# OpenReview forum: "MetaPhys: Few-Shot Adaptation for Non-Contact Physiological Measurement"
_ICLR.cc/2021/Conference — Reject_

### Official Review · AnonReviewer4 · 2020-10-21
**Recommendation to Reject**

**Rating:** 4
**Confidence:** 3

**Review:**

Summary:
The paper proposes an (un)supervised meta-learning framework for few-shot adaptation in camera-based vital sign measurements denoted as "MetaPhys". The goal is to develop few-shot adapting personalized frameworks for remote physiological measurements with limited data. Authors perform cross-dataset evaluations to illustrate the model's few-shot adaptation capabilities.

Positives:
The authors present their application domain and purpose very clearly, and the related work from literature are nicely explored. Experimental evaluations are thorough and the comparisons to state-of-the-art unsupervised approaches are done well.

Concerns:
Unfortunately I do not see any methodological contribution in this paper from a "representation learning" perspective. MetaPhys is (as far as it is presented in the paper) a framework on using model-agnostic meta-learning (MAML) on physiological measurement data in unsupervised subject adaptation settings. Since MAML is already a model-agnostic approach, the paper seems to me like a well-presented reformulation and implementation of a well-known meta-learning algorithm in this respective application domain. The paper does not make a clearly stated distinction of their approach/contribution from a representation learning standpoint.
The authors list their contributions in four bullet-points, where the third and fourth ones relate to analyses of temporal and appearance features during model personalization, and analyses for subjects with different skin types. However these analyses are only addressed superficially (not described or analyzed in detail) at the end of the paper. Hence I would be questioning these as "contributions" of this paper.

Minor comment:
In fact, the authors present MetaPhys for both supervised and unsupervised few-shot adaptation settings. Then why did the authors use "unsupervised" indication for the paper's title (and even in Abstract)?

---

> ### Author Response · Authors · 2020-11-14
> **Responses about ML contribution an novelty**
>
> Thanks for your great comments. First, meta learning / few shot learning is usually supervised and we argue that unsupervised and weakly supervised approaches are very useful. Second, in this work, we advance the field by leveraging traditional unsupervised learning techniques (i.e., in our case, POS) to provide pseudo labels in a state-of-the-art meta learning framework to train MetaPhys. We are the first to propose an unsupervised deep learning method in the domain of physiological sensing, other unsupervised methods have relied on traditional signal processing and de-mixing (e.g., independent component analysis). Although TS--CAN, MAML and POS do exist, we are the first to integrate and use those three different components in a novel way and our results show a considerable improvement over the state-of-the-art.
>
> In this work, we provided a certain degree of explanation for why MetaPhys outperforms existing methods, highlighting that both appearance and motion (pixel difference) information is important in the adaptation phase. To our best knowledge, we are the first providing insights why meta learning helps achieve better results in remote physiological sensing. However, we appreciate that we could temper our statements in the introduction and have edited that text. We have changed the third contribution to be: “To perform an ablation experiment, freezing weights in the temporal and appearance branches to test sensitivity during adaptation.” We feel this is more representative of our experiment.
>
> We believe our analysis using meta learning to personalize remote physiological sensing is particularly important for designing approaches that are robust to skin tone and note that bias is a particular problem in optical measurement.
>
> We agree with the reviewer that the title is confusing, and we have updated it to “MetaPhys: Few-Shot Adaptation for Non-Contact Physiological Measurement”.

---

> > ### Comment · AnonReviewer4 · 2020-11-19
> > **Reply to authors**
> >
> > Thanks to the authors for their time and responses.
> >
> > I would not fully agree on the authors' statement that "meta learning is usually supervised". There exists a line of intuitive work from the past few years for various unsupervised approaches to this problem (e.g., self-supervised training, meta pseudo-labeling). However I certainly agree on the authors' statement that unsupervised meta learning is fundamentally more beneficial in personalized healthcare applications.
> >
> > From the perspective of an ICLR audience, I can not tell a clear relevant novelty in the current study. The authors present a powerful integration of several state-of-the-art methods (mainly harnessing MAML) in a new application domain. I do not believe that the technical novelty of this study is presenting an advancement in machine learning. Hence I would still suggest that the current study would be more suitable for a biomedical sciences/engineering audience.

---

### Official Review · AnonReviewer1 · 2020-11-01
**A medical domain application that demonstrates that meta learning is useful**

**Rating:** 5
**Confidence:** 3

**Review:**

Contribution-
In this paper, the authors present a few-shot learning model for non-contact physiological signal measurement to build a more accurate and convenient personalized health sensing system. The motivation is that traditional fine-tuning method for this task is difficult since it requires large sets of high-quality training data for specific individuals, due to differences between each individual, measurement environment, and camera sensor condition. Therefore, the authors applied MAML on top of the existing deep learning network (TS-CAN) and implemented a model that aims to learn fast from a small number of training samples. The main contributions of this paper are: a meta-learning model that supports both supervised and unsupervised few-shot adaptation; improved performance by about 40% compared to a baseline that does not use meta learning; empirical analysis of performance for subjects with different skin types.

Strengths-
Few-shot learning could be beneficial for remote patient monitoring applications.
Experimental results on all three datasets are positive, with around 40%-50% MAE improvement compared to the baseline.
Provided experimental evaluation provides some insight into why the proposed approach works well in this application.

Weaknesses-
This is a straightforward application of meta learning / few shot learning to a specific domain. Other than showing that meta learning / few shot learning is useful in yet another particular application, it is not clear how is this work advancing machine learning.
Some explanations could have been more specific. For example, it is not clear how exactly are ‘pseudo labels’ generated and how do they compare with the ground truth labels.
The experimental results do not do enough to provide some novel insight into MAML. For example, it would be good to see a comparison between pretrained model and MAML model performance as a function of data size, and how robust to hyperparameters are the competing approaches.

Summary of the rating:
While the experimental results are positive, there is little methodological novelty and the experimental results do not seem to provide some interesting insights that might spur machine learning research in some novel directions.

---

> ### Author Response · Authors · 2020-11-14
> **Responses about novelty and method**
>
> Thanks for your insightful feedback. We’d like to highlight the novelty in our work. Meta learning/few shot learning is usually conducted in a supervised manner and there is still relatively little work exploring unsupervised methods/performance. In this work, we leverage an optically-grounded unsupervised learning technique and a modern meta learning framework to jointly train MetaPhys. This is also the first work that leverages pseudo labels in training a physiological sensing model and the first unsupervised deep learning method in remote physiological measurement. Our results indicate that even a relatively simple de-mixing approaches provides labels that can be used in a meta learning context to obtain strong results and that training with these, sometimes noisy labels, well outperforms the de-mixing approach itself.
>
> In Table 1, we show the results from POS, MetaPhys improves the performance (reducing MAE by 53% compared to POS alone). In terms of how POS works, it can be summarized in four main steps: 1) spatial averaging of each video frame, 2)  temporal normalization, 3) de-mixing/source signal separation using a fixed matrix projection to remove specular reflections and other noise, 4) band-pass filtering. We have added more details about POS in the paper.
>
> We have included more detailed results in the Section 5 about comparison between pretrained and MAML model performance as a function of  size of the support set, and robustness to hyperparameters. Specifically, we experimented with different window sizes in MetaPhys as we think it is the most important parameter that could be tuned.  We tested providing a 6s, 12s and 18s window of data for model adaptation (i.e., the support set) during meta training and testing, and the results showed training with 18s (RMSE: 3.12 BPM) outperformed 6s (RMSE: 5.43) and 12s (RMSE: 5.53) in the MMSE dataset. We found a similar trend on the UBFC dataset as well. (RMSE of 18s: 3.12, RMSE of 12s: 4.48, RMSE of 6s: 3.46).  There are not many other hyperparameters to examine, but if you would like us to present results with different learning rates or other parameters we would be happy to include them.

---

### Official Review · AnonReviewer5 · 2020-11-06
**The paper introduce a method for performing person-specific physiological sensing from videos.**

**Rating:** 6
**Confidence:** 5

**Review:**

The authors propose a system called MetaPhys for personalized remote physiological sensing from videos.  Their system combines a pre-trained CNN with an existing meta learning method (MAML). The investigated both supervised and unsupervised training of their system.  Performance evaluation of their methods on benchmark datasets show their model significantly outperforms SOTA methods using multiple metrics as well as for different skin types. They further show that the unsupervised model achieves comparable results to the supervised model.

This is an interesting paper, with extensive evaluation that demonstrates their model’s superior performance  to SOTA methods. The technical novelties of their model, however, are incremental since it is primarily built the existing models : a pre-trained CNN, an existing meta-learning method (MAML), and use of the POS method to generate pseudo-labels for unsupervised learning.

---

> ### Author Response · Authors · 2020-11-14
> **Clarification of our contribution**
>
> Thank you for your helpful and constructive comments. We would like to argue that although the pre-trained TS--CAN, MAML and POS exist, we are the first to integrate and use those three different components in a novel way. Moreover, we are the first to propose and evaluate an unsupervised deep learning architecture in the field of remote physiological sensing. Even though the unsupervised approach is simple, it still substantially improves the state-of-the-art results and also doesn't bring additional computational burden. Given the utility of unsupervised methods where personalization is important, as is the case here, and that this is a task with positive applications in healthcare we feel that our paper is a valuable contribution.

---

> > ### Comment · AnonReviewer5 · 2020-11-18
> > **Reply to the authors rebuttal**
> >
> > I agree with the authors on the application significance and performance improvements over existing methods. But I have to say technical novelty remains incremental; both the integration and unsupervised learning represent marginal technical contributions.

---

### Decision · Program_Chairs · 2021-01-07
**Final Decision**

**Decision:**

Reject

**Comment:**

This application-oriented paper has been carefully evaluated by three expert reviewers. Their assessments all agreed on a quite marginal methodological novelty of the presented work, yet they recognized nicely engineered pipeline of pre-existing modules that appears to satisfy an important remote sensing application. I agree with that assessment and concur with the reviewer's opinion that the work as presented, in spite of being of potential interest to healthcare or biomedical communities, will be of little interest to the ICLR audience. Therefore I recommend rejecting it.